# Pemafibrate Induces a Low Level of PPARα Agonist-Stimulated mRNA Expression of ANGPTL4 in ARPE19 Cell

**DOI:** 10.3390/bioengineering11121247

**Published:** 2024-12-09

**Authors:** Hiroshi Ohguro, Nami Nishikiori, Tatsuya Sato, Megumi Watanabe, Megumi Higashide, Masato Furuhashi

**Affiliations:** 1Departments of Ophthalmology, School of Medicine, Sapporo Medical University, S1W17, Chuo-ku, Sapporo 060-8556, Japan; nami076@yahoo.co.jp (N.N.); watanabe@sapmed.ac.jp (M.W.); megumi.h@sapmed.ac.jp (M.H.); 2Departments of Cardiovascular, Renal and Metabolic Medicine, Sapporo Medical University, S1W17, Chuo-ku, Sapporo 060-8556, Japan; satatsu.bear@gmail.com; 3Departments of Cellular Physiology and Signal Transduction, Sapporo Medical University, S1W17, Chuo-ku, Sapporo 060-8556, Japan

**Keywords:** intraocular cells, lipid metabolism, PPARα, PPARγ, GRP41, GRP42, extracellular flux analyzer

## Abstract

To elucidate the unidentified roles of a selective peroxisome proliferator-activated receptor α (PPARα) agonist, pemafibrate (Pema), on the pathogenesis of retinal ischemic diseases (RID)s, the pharmacological effects of Pema on the retinal pigment epithelium (RPE), which is involved in the pathogenesis of RID, were compared with the pharmacological effects of the non-fibrate PPARα agonist GW7647 (GW). For this purpose, the human RPE cell line ARPE19 that was untreated (NT) or treated with Pema or GW was subjected to Seahorse cellular metabolic analysis and RNA sequencing analysis. Real-time cellular metabolic function analysis revealed that pharmacological effects of the PPARα agonist actions on essential metabolic functions in RPE cells were substantially different between Pema-treated cells and GW-treated cells. RNA sequencing analysis revealed the following differentially expressed genes (DEGs): (1) NT vs. Pema-treated cells, 37 substantially upregulated and 72 substantially downregulated DEGs; (2) NT vs. GW-treated cells, 32 substantially upregulated and 54 substantially downregulated DEGs; and (3) Pema vs. GW, 67 substantially upregulated and 51 markedly downregulated DEGs. Gene ontology (GO) analysis and ingenuity pathway analysis (IPA) showed several overlaps or differences in biological functions and pathways estimated by the DEGs between NT and Pema-treated cells and between NT and GW-treated cells, presumably due to common PPARα agonist actions or unspecific off-target effects to each. For further estimation, overlaps of DEGs among different pairs of comparisons (NT vs. Pema, NT vs. GW, and Pema vs. GW) were listed up. Angiopoietin-like 4 (ANGPTL4), which has been shown to cause deterioration of RID, was the only DEG identified as a common significantly upregulated DEG in all three pairs of comparisons, suggesting that ANGPTL4 was upregulated by the PPARα agonist action but that its levels were substantially lower in Pema-treated cells than in GW-treated cells. In qPCR analysis, such lower efficacy for upregulation of the mRNA expression of ANGPTL4 by Pema than by GW was confirmed, in addition to substantial upregulation of the mRNA expression of HIF1α by both agonists. However, different Pema and GW-induced effects on mRNA expression of HIF1α (Pema, no change; GW, significantly downregulated) and mRNA expression of ANGPTL4 (Pema, significantly upregulated; GW, significantly downregulated) were observed in HepG2 cells, a human hepatocyte cell line. The results of this study suggest that actions of the PPARα agonists Pema and GW are significantly organ-specific and that lower upregulation of mRNA expression of the DR-worsening factor ANGPTL4 by Pema than by GW in ARPE19 cells may minimize the risk for development of RID.

## 1. Introduction

Ligand-activated transcription factors belonging to the superfamily of nuclear receptors called peroxisome proliferator-activated receptors (PPARs), which include three isoforms (PPARα, PPARγ, and PPARβ/δ), function to regulate various genes related to physiological energy production, glucose, and lipid metabolism [1,2]. PPARs are involved in the etiology of various diseases and disorders, including metabolic syndrome [3], type 2 diabetes mellitus (DM) [2,4], cancer, non-alcoholic fatty liver diseases [5], cardiovascular diseases [6], and neurological disorders [1]. In ocular pathogenesis, PPARs also play pivotal roles in retinal ischemic diseases (RIDs) such as diabetic retinopathy (DR) and age-related macular degeneration (AMD) and in other ocular diseases, including optic neuropathy and dry eye [7]. PPARs are thus recognized as appropriate therapeutic targets for these diseases, and, in fact, various modulators for PPARs have been clinically used [1].

The selective PPARα modulator pemafibrate (Pema) has been shown to have a very potent PPARα agonist activity, and very high PPARα selectivity has been shown to not only improve serum lipid levels as other fibrates do but also minimize various other fibrate-induced off-target effects, such as liver and kidney dysfunctions, as compared to other fibrates, such as fenofibrate and bezafibrate [8,9,10]. Although a recent randomized controlled trial did not show that Pema effectively reduced cardiovascular risk by lowering serum triglyceride levels [11], data for fibrate-induced reduction of cardiovascular risk were shown by meta-analyses of patients with high TG levels or mixed dyslipidemia (elevated TG and low HDL-C) [12]. In addition to the preferable outcomes in therapy for systemic metabolic diseases, it has been suggested that Pema may also be applicable as one of the therapeutic options for preventing RID [13]. In fact, previous studies showed that Pema could also suppress retinal dysfunction in murine models with streptozotocin-induced DR [14,15]. Collectively, these observations strongly suggested that the PPARα agonist action of Pema may have beneficial effects for preventing RID. However, at the time of writing paper, the difference between the intraocular PPARα agonist action of Pema and that of other non-fibrate PPARα agonists has not been studied, especially in the retinal pigment epithelium (RPE), which is one of the main responsible intraocular segments to the pathogenesis of RID [16].

Therefore, the present study was carried out to elucidate the unidentified pharmacological differences between the intraocular PPARα agonist action of Pema and that of the non-fibrate PPARα agonist GW7647 (GW) on cellular metabolic functions via analysis using a Seahorse Bioanalyzer with ARPE19 cells, which originate from the RPE. In addition, to elucidate the possible underlying molecular mechanisms causing these differences in pharmacological effects, RNA sequencing analysis was carried out using Pema-treated and GW-treated ARPE19 cells.

## 2. Materials and Methods

### 2.1. Planar Cultures of ARPE19 Cells

All experiments using human-derived cells, including commercially available human retinal pigment epithelium cells (ARPE19 cells, ATCC, and #CRL-2302™) and human hepatocytes (HepG2 cells, ATCC, and #HB-8065), were conducted in compliance with the tenets of the Declaration of Helsinki after approval by the internal review board of Sapporo Medical University. ARPE19 cells and HepG2 cells were cultured in HG-DMEM supplemented with 10% FBS, 1% L-glutamine, and 1% antibiotic–antimycotic and were maintained by changing the medium daily under standard normoxia conditions (37 °C, 20% O_2_/5% CO_2_).

### 2.2. Gene Expression Analyses

Extraction of total RNA following reverse transcription and quantitative real-time PCR (qRT-PCR) was carried out as previously reported [17,18], using specific primers and probes (Appendix A). Normalization of each respective gene expression was compared with the expression of internal control, 36B4 (Rplp0).

### 2.3. Real-Time Measurements of Cellular Metabolic Functions

Measurements of oxygen consumption rate (OCR) and extracellular acidification rate (ECAR) of planar cultured ARPE19 cells that were treated with or not treated with lipid modulators were carried out using a Seahorse XFe96 Bioanalyzer (Agilent Technologies, Santa Clara, CA, USA), as described in our previous report [19]. Prior to those measurements, the cells were pre-incubated for 24 h with various lipid metabolism modulators toward PPARα agonists, 10 μM Pema [15] and 20 μM GW [20,21], and those concentrations were confirmed in previous studies to be optimum concentrations. In addition, both drugs induced maximal human PPARα activity and had similar potencies (EC50 of 1.8 × 10^−11^ M for GW compared to EC50 of 2.3 × 10^−11^ M for Pema) [22].

Metabolic indices were calculated as follows: basal respiration, subtraction of OCR with rotenone/antimycin A from OCR at baseline; ATP-linked respiration, the difference in OCR after the addition of oligomycin; proton leak, subtraction of OCR with rotenone/antimycin A from OCR after the addition of oligomycin; maximal respiration, subtraction of OCR with rotenone/antimycin A from OCR after the addition of FCCP; spare respiratory capacity, subtraction of OCR at baseline from OCR after the addition of FCCP; non-mitochondrial respiration, OCR with rotenone/antimycin A; basal ECAR, subtraction of the end point of ECAR after the injection of 2-DG from ECAR at baseline; glycolytic capacity, subtraction of the end point of ECAR after the injection of 2-DG from ECAR with oligomycin; glycolytic reserve, subtraction of ECAR at baseline from ECAR with oligomycin; non-glycolytic acidification, the end point of ECAR after injection of 2-DG; and baseline OCR/ECAR ratio, division of OCR at baseline by ECAR at baseline.

### 2.4. RNA Sequencing Analysis of Gene Functions and Analysis of Pathways

Total RNA was extracted from 2D confluent cells of ARPE19 cells that were untreated or treated with 10 μM Pema or 20 μM GW for 24 h in a 150 mm dish, using an RNeasy mini kit (Qiagen, Valencia, CA, USA), and then RNA extraction and next-generation sequencing were performed as described recently [23]. Genes with log2 fold-change ≥ ±1 and false discovery rate (FDR)-adjusted *p*-value < 0.05 and *q* < 0.08 between groups were determined as differentially expressed genes (DEGs). Various analyses to predict various pathways and networks, including predicting possible upstream regulators and causal network regulators, were performed by ingenuity pathway analysis (IPA, Qiagen, https://www.qiagenbioinformatics.com/products/ingenuity-pathway-analysis, the accessed date 19 June 2024) [24], as shown in recent studies [24,25,26].

### 2.5. Statistical Analysis

All statistical analyses were performed using Graph Pad Prism version 9 (GraphPad Software, San Diego, CA), as described in our recent report [17,18]. In brief, one-way ANOVA was used to determine statistical significance for multiple groups, and significant one-way ANOVA was followed by Tukey’s HSD (Honestly Significant Difference) post hoc analysis. A *p*-value less than 0.05 was statistically significant, as indicated by asterisks.

## 3. Results

As a possible working hypothesis in the present study, we expected to detect some novel different mechanisms related to the pathogenesis of RID between Pema and GW in several biological analyses despite their identical activities for stimulating PPARα. To elucidate such possible mechanisms of Pema, the pharmacological effects of Pema on cells involved in the pathogenesis of RID were compared with the pharmacological effects of another non-fibrate PPARα agonist, GW, by using cellular metabolic function analysis and RNA sequencing analysis. In a previous study using COS-7 cells, Pema and GW had similar EC50 values of 2.3×10^−11^ M and 1.8×10^−11^ M, respectively [22]. Another recent study using a coactivator recruitment assay for PPARα showed that approximately 10^−6^ M of Pema reached the level of maximum activities of PPARα activation by GW [27]. In addition to these results, appropriate concentrations of the two drugs for stimulating PPARα in RPE cells may be different from the concentrations in those studies, and there has been no previous study using these drugs in RPE cells. However, in previous studies using neuronal cells, (1) 1 μM to 100 μM of GW was used in a cell culture experiment [21,28], and (2) PPARα stimulation activities of Pema were higher in concentration response curves [22]. Considering those collective results, 10 μM of Pema and 20 μM of GW, for which maximum and identical PPARα stimulation should be induced, were used in the following analyses.

Initially, the effects of Pema (10 μM) on glycolysis and mitochondrial functions of ARPE19 cells determined by Seahorse cellular metabolic function analysis were compared with the effects of GW (20 μM). As shown in Figure 1, Pema and GW7647 had significantly different effects on cellular metabolic functions despite the fact that they are both PPARα agonists. Although ATP-linked respiration was increased by both Pema and GW, maximal respiration and spare respiration were significantly decreased only by GW. Also, glycolytic capacity was increased by Pema but not by GW. Furthermore, the baseline OCR/ECAR index was significantly increased by GW compared with the effect of Pema in ARPE19 cells. These results suggested that the pharmacological effects of various PPARα agonists on essential metabolic functions in RPE cells are substantially different.

To elucidate the possible molecular mechanisms underlying the different effects of Pema and GW on cellular metabolic functions in ARPE19 cells, RNA sequencing analysis was carried out using non-treated ARPE19 cells, Pema-treated ARPE19 cells, and GW-treated ARPE19 cells (n = 3 each). In a heatmap (Figure 2) and an M-A plot (Appendix A) and a volcano plot (Appendix A), gene expression profiles in these three conditions of ARPE19 cells were significantly different. The identified differentially expressed genes (DEGs) included 37 markedly upregulated and 72 markedly downregulated DEGs between non-treated ARPE19 cells and Pema-treated ARPE19 cells (NT vs. Pema), 32 markedly upregulated and 54 markedly downregulated DEGs between non-treated ARPE19 cells and GW-treated ARPE19 cells (NT vs. GW), and 67 markedly upregulated and 51 markedly downregulated DEGs between Pema-treated ARPE19 cells and GW-treated ARPE19 cells (Pema vs. GW) (a list of all of the DEGs is included in the Appendix A). Using GO enrichment analysis of those DEGs, three GO term-related biological processes between NT vs. Pema and NT vs. GW were compared. The top 3 of numbers of gene counts in the cellular component (Appendix A, NT vs. Pema; and Appendix A, NT vs. GW) in both conditions were (1) membrane, (2) plasma membrane, and (3) extracellular region. Those in the molecular function (Appendix A, NT vs. Pema; and Appendix A, NT vs. GW) of both conditions were (1) calcium ion binding, (2) signaling receptor binding, and (3) serine-type endopeptidase activities. Those in the biological process in NT vs. Pema (Appendix A) and NT vs. GW (Appendix A) were (1) inflammatory response, (2) potassium ion transmembrane transport, and (3) nucleosome assembly; and (1) signal transduction, (2) cell adhesion, and (3) hemophilic cell adhesion via plasma membrane adhesion molecules, respectively. Furthermore, IPA analysis showed that some overlap issues were identified between NT vs. Pema and NT vs. GW in the top five networks (Table 1), including connective tissue disorders, developmental disorders, cancer, and organismal injury and abnormalities; and in the top five canonical pathways (Table 2), including class A/1 (rhodopsin-like receptors) and Gα signaling events and G protein-coupled receptor signaling. These collective observations suggested that Pema and GW had both common biological activities presumably as PPARα agonists and distinct biological activities that were independent of PPARα signaling.

Next, to elucidate further the unidentified mechanisms related to and not related to PPARα stimulation, commonly detected DEGs among upregulated and downregulated DEGs observed between NT vs. Pema, NT vs. GW, and Pema vs. GW were listed. As shown in Table 3 and Figure 3, these commonly observed DEGs were categorized into the following four groups.

(1)Group 1: Four downregulated DEGs, namely histone H1.5 (H1-5), NPEPPS pseudo 1 (NPEPPSP1), inhibitor of nuclear factor kappa B kinase subunit gamma pseudogene 1 (IKBKGP1), and Actin-Like 10 (ACTL10); and two upregulated DEGs, namely polycystic kidney disease 1 pseudogene 1 (PKD1P1) and septin 7 pseudogene 3 (SEPTIN7P3), were commonly observed in NT vs. Pema and NT vs. GW, and they were thus most likely to be related to PPARα stimulation in ARPE19 cells.(2)Group 2: Four downregulated DEGs, namely EF-hand domain containing 2 (EFHC2), FERM and PDZ domain containing 2B pseudogene (FRMPD2B), LINC00910, and phosphodiesterase 7B (PDE7B); and six upregulated DEGs, namely hematopoietic cell-specific Lyn substrate 1 (HCLS1), farnesyl diphosphate synthase pseudogene 2 (FDPSP2), exocyst complex component 5 pseudogene 1 (EXOC5P1), Solute Carrier Family 4 Member 1 Adaptor Protein Pseudogene 1 (SLC4A1APP1), TRPM8 channel-associated factor 2 pseudogene 1 (TCAF2P1), and MIR193BHG, were commonly observed in NT vs. GW and Pema vs. GW, and they were thus related to GW-related specific functions unrelated to PPARα stimulation.(3)Group 3: Three DEGs, namely small nucleolar RNA H/ACA box 66 (SNORA66), zinc finger protein 890 pseudogene (ZNF890P), and small integral membrane protein 11 (SMIM11), were downregulated in comparison of GW vs. Pema and upregulated in comparison of NT vs. Pema, and these genes were thus thought to be related to Pema-related specific functions unrelated to PPARα stimulation.(4)Group 4: One DEG, angiopoietin-like 4 (ANGPTL4), was commonly observed as an upregulated DEG in all three comparisons Therefore, upregulation of this gene was induced by both Pema and GW via their PPARα stimulatory activities, with the stimulatory activity of GW being more potent than that of Pema.

The gene expression of ANGPTL4 and hypoxia-inducible factor 1a (HIF1α) in ARPE19 cells and HepG2 cells in the presence or absence of the PPARα agonist Pema or GW was examined by qPCR analysis (Figure 4). The PPARα agonists caused significant upregulation of the gene expression of ANGPL4 in ARPE19 cells in a dose-dependent manner, with the effect of GW being much more potent than that of Pema. Similarly, the gene expression of HIF1α was also relatively increased in a dose-dependent manner, and levels of those were significantly lower in Pema compared to GW. However, in HepG2 cells, the mRNA expression of ANGPTL4 and HIF1α was not significantly altered by 1nM~10 μM Pema or GW, except for a marked upregulation of ANGPTL4 by 10 μM Pema. Collectively, the results suggested that PPARα agonist-induced effects by Pema and GW may be exclusively organ-specific and that the lower efficacy of Pema for upregulation of the mRNA expression of ANGPTL4 may reduce the risk for RID pathogenesis.

## 4. Discussion

The family of proteins ANGPTL 1~8, which have structural similarity with angiopoietins (ANGPTs), play various physiological roles in the regulation of lipid and glucose metabolism and hematopoietic stem cell expansion and pathological roles related to chronic inflammation, angiogenesis, and vascular permeability [1,2,3,4,29,30,31,32]. Among the ANGPTL family proteins, ANGPTL4 was simultaneously identified as a target for PPARα and PPARγ, a fasting-induced factor from the liver, and an angiopoietin-related protein [33,34,35,36]. ANGPTL4 is preferentially expressed in adipose tissues and the liver, with less expression in other tissues, and its expression is critically regulated by several transcription factors, including PPARs, glucocorticoid receptors, and HIF1α in response to different nutritional and metabolic conditions [34,35,37,38]. In the ophthalmology field, ANGPTL4 was first shown to be a possible vasoactive cytokine to facilitate vascular permeability and macular edema in patients with DR [39]. It was shown that aqueous levels of ANGPTL4 in patients with diabetic macular edema were significantly higher than those in non-DM patients with cataracts [40,41], and it was also demonstrated that vitreous levels of ANGPTL4 were elevated in patients with proliferative DR and that ANGPTL4 levels were substantially correlated with levels of serum lipids [42]. A previous study using human retinal microvascular endothelial cells and diabetic rats suggested that activation of ANGPTL4 was exclusively dependent on overexpression of HIF1α under high-glucose conditions [43]. Furthermore, it was shown by using ARPE19 cells and a DM rat model that ANGPTL4 induced a barrier function of the RPE by activating STAT3 [44,45]. Collectively, the results rationally suggested that ANGPTL4 may be a risk factor for DR and thus could be used as a potential therapeutic target for treatment of DR. In the current study, we first showed the different efficacy of Pema and GW for upregulating ANGPTL4 in ARPE19 cells: Pema-induced upregulation of ANGPTL4 was significantly less than that of GW, suggesting that Pema is the preferable PPARα agonist to minimize ANGPTL4-related retinal dysfunction.

In contrast to ARPE19 cells, mRNA expression of ANGPTL4 was also significantly upregulated by Pema but was markedly downregulated by GW in HepG2 cells. Such Pema-induced upregulation of the gene expression of ANGPL4 in human hepatocyte cells, including HepG2 cells, was also shown in a previous study [46]. However, in contrast, another study showed that ANGPTL4 expression in HepG2 cells was activated by the PPARδ activator GW501516 but not by the PPARα agonist Wy-14643 [47]. Since we observed that GW decreased the mRNA expression level of ANGPTL4 in HepG2 cells but increased it in ARPE19 cells, these diverse effects may be caused by organ-specific PPARα agonist-induced effects and/or some unknown off-target effects that are different between Pema and GW on ANGPTL4 expression. In addition, our results showing that mRNA expression of HIF1α similarly fluctuated as ANGPTL4 in both ARPE19 cells and HepG2 cells rationally supported previous observations that ANGPLT4 was transcriptionally regulated by HIF1α immediately after hypoxic stimulation [48,49]. In addition, a previous study showed that overexpression of ANGPTL4 deteriorates mitochondrial functions in the liver tissues of db/db diabetic mice by downregulating numerous proteins associated with mitochondrial respiration [50]. Consistent with this report, the present study demonstrated that the mitochondrial respiratory capacity in ARPE19 cells was significantly small from treatment with GW compared to Pema, as shown in Figure 1, presumably due to higher upregulation of ANGPTL4 by GW. Furthermore, the baseline OCR/ECAR ratio suggested that metabolic dominance in glycolysis compared to mitochondrial function was more evident in the order of non-treated control, Pema, and GW. Since the levels of mRNA expression of ANGPTL4 increased in that order, we speculated that ANGPTL4 may be critically involved in the regulation of cellular metabolic functions in ARPE19 cells. Indeed, previous studies showed that ANGPTL4 enhances glycolysis in cancerous cells [51,52]. Taken together, the excessive upregulation of ANGPTL4 in retinal pigment epithelial cells may be one of the pivotal mechanisms that increase the risk of retinal dysfunction via metabolic impairments.

We acknowledge that the present study has several limitations. Firstly, the mechanisms underlying the different effects of Pema and GW on mRNA expression of ANGPTL4 and HIF1α, as well as the difference in their effects in ARPE19 cells and HepG2 cells, have not been elucidated, despite the fact that Pema and GW were used under conditions of maximal and identical PPARα stimulation. This suggested that some unidentified off-target effects specific to Pema but not GW may modulate PPARα-induced mRNA expression of ANGPTL4 and HIF1α. In fact, several DEGs in NT vs. Pema, but not NT vs. GW, were identified in the Appendix A, and therefore an additional study is required to re-analyze the RNA-seq data. Secondly, several other identified genes that were shown to be specific to Pema or GW7467 by IPA analysis were not investigated; however, most of those were pseudogenes, non-protein coding RNA, or small nuclear RNA. Thirdly, other fibrates, including fenofibrate and bezafibrate, that are currently used for patients were not investigated. Fourthly, ARPE19 cells used in the present study may have biological aspects that are different from those of in vivo RPE cells. Fifthly, it has been shown that ANGPTL4 forms an oligomer and is processed by proteolytic cleavage to exert its biological activities [29]. Therefore, additional studies are required (1) to elucidate unidentified mechanisms of Pema and GW by which diverse gene expression of ANGPTL4 and HIF1α is induced; (2) to identify the molecular mechanisms by which diverse metabolic changes among PPARα agonists, including other fibrates, are induced; and (3) to determine biological aspects of ANGPTL4 at protein levels for a deeper understanding of the pharmacological roles of PPARα agonists, including Pema, in RID pathogenesis and for application of the agonists as possible therapeutic strategies for various RID using various primary cultured RPE cells, as well as in vivo rodent models with RID.

## Figures and Tables

**Figure 1 bioengineering-11-01247-f001:**
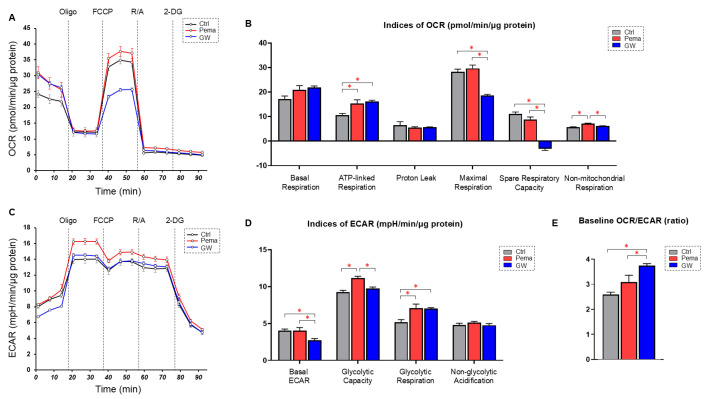
Effects of the PPARα agonists pemafibrate (Pema) and GW7647 (GW) on metabolism in ARPE19 cells. ARPE19 cells that were not treated with (Ctrl) and those that were treated with Pema or GW were subjected to a Seahorse real-time metabolic function analysis. Measurement of oxygen consumption rate (OCR, panel (**A**)). Measurement of extracellular acidification rate (ECAR, panel (**B**)). Indices of mitochondrial function (panel (**C**)). Indices of glycolytic function (panel (**D**)). Baseline OCR/ECAR ratio (panel (**E**)). Freshly prepared specimens were used in all experiments (n = 6). Data are shown as means ± standard error of the mean (SEM). * *p* < 0.05. Oligo, oligomycin; FCCP, carbonyl cyanide p-trifluoromethoxyphenylhydrazone; R/A, rotenone/antimycin A; 2-DG, 2-deoxyglucose.

**Figure 2 bioengineering-11-01247-f002:**
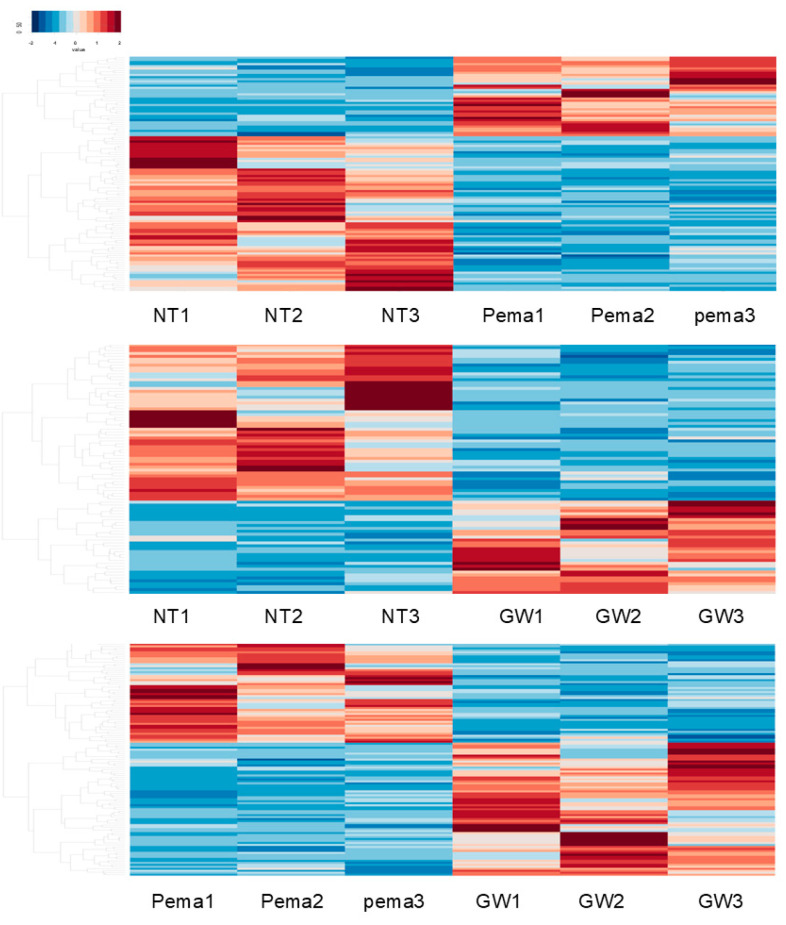
Heatmaps for DEGs in ARPE19 cells in three comparison conditions: not treated with a PPARα agonist (NT 1-3) vs. treated with Pema 1-3 ((**upper**) panel), not treated with a PPARα agonist (NT 1-3) vs. treated with GW 1-3 ((**middle**) panel), and treated with Pema 1-3 vs. treated with GW 1-3 ((**lower**) panel). Two-dimensionally cultured ARPE19 cells not treated with a PPARα agonist (NT, n = 3) and those treated with 10 μM of Pema (n = 3) or 20 μM of GW (n = 3) were subjected to RNA sequencing analysis. A hierarchical clustering heatmap of differentially expressed genes (DEGs) is shown. Either overexpressed (red) or underexpressed (blue) DEGs in NT cells compared with those in Pema cells are shown in the (**upper**) panel. Either overexpressed (red) or underexpressed (blue) DEGs in NT cells compared with those in GW cells are shown in the (**middle**) panel. Either overexpressed (red) or underexpressed (blue) DEGs in Pema cells compared with those in GW cells are shown in the (**lower**) panel.

**Figure 3 bioengineering-11-01247-f003:**
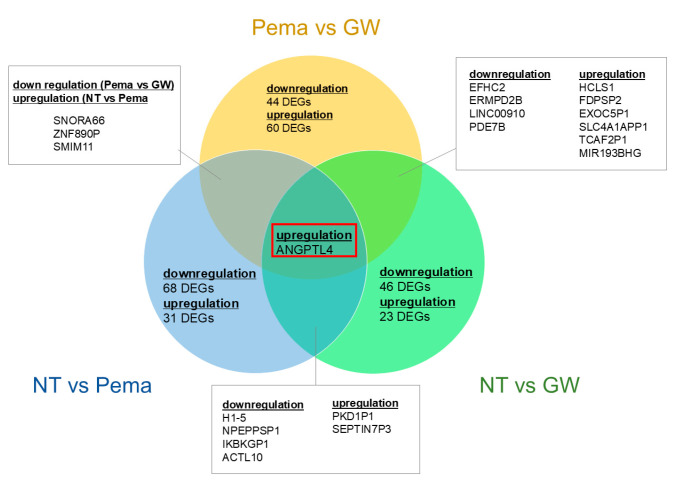
Venn diagram to represent DEGs detected in different categories. DEGs obtained from three different comparisons, namely NT vs. Pema, NT vs. GW, and Pema vs. GW, are shown in a Venn diagram. Gene names are listed in the overlapping area of each circle. ANGPTL4 was commonly observed as an upregulated DEG in these three comparisons. H1-5, histone H1.5; ACTL10, Actin-Like 10; EFHC2, EF-hand domain containing 2; LINC00910, Long Intergenic Non-Protein Coding RNA 910; PDE7B, phosphodiesterase 7B; SNORA66, small nucleolar RNA H/ACA box 66; SMIM11, small integral membrane protein 11; HCLS1, hematopoietic cell-specific Lyn substrate 1; MIR193BHG; MIR193b-365a host gene; ANGPTL4, angiopoietin-like 4; IKBKGP1, Inhibitor of nuclear factor kappa B kinase subunit gamma pseudogene 1; PKD1P1, polycystic kidney disease 1 pseudogene 1; SEPTIN7P3, septin 7 pseudogene 3; FRMPD2B, farnesyl diphosphate synthase pseudogene 2; EXOC5P1, exocyst complex component 5 pseudogene 1; SLC4A1APP1, Solute Carrier Family 4 Member 1 Adaptor Protein Pseudogene 1; TCAF2P1, TRPM8 channel-associated factor 2 pseudogene 1; ZNF890P, zinc finger protein 890 pseudogene.

**Figure 4 bioengineering-11-01247-f004:**
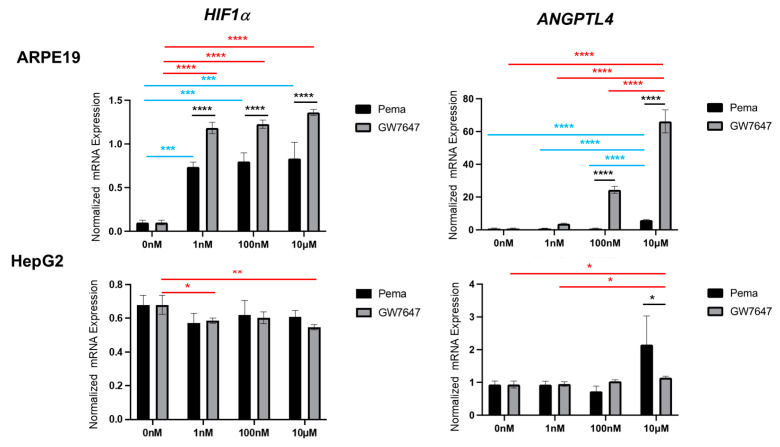
qPCR analysis for ANGPTL4 and HIF1α in PPARα agonist-treated ARPE19 cells and HepG2 cells. Two-dimensionally cultured ARPE19 cells and HepG2 cells not treated or treated with a PPARα agonist, GW or Pema, were subjected to qPCR analysis, and the mRNA expression of *ANGPTL4* and *HIF1α* was estimated. Experiments were repeated three times, using freshly prepared cells (n = 3 each), in each experiment. * *p* < 0.05, **** *p* < 0.001 (Pema vs. GW), * *p* < 0.05, ** *p* < 0.01, **** *p* < 0.001 (among different concentrations of GW), *** *p* < 0.005, and **** *p* < 0.001 (among different concentrations of Pema).

**Table 1 bioengineering-11-01247-t001:** Top 5 of networks.

**NT vs. Pema**		
**ID**	**Associated Network Functions**	**Score**
1	Connective tissue disorders, developmental disorder, hereditary disorder	47
2	Developmental disorder, hereditary disorder, organismal injury and abnormalities	40
3	Developmental disorder, endocrine system disorders, hereditary disorder	36
4	Cancer, neurological disease, organismal injury and abnormalities	32
5	Cell-to-cell signaling and interaction, cellular response to therapeutics	32
**NT vs. GW**		
**ID**	**Associated Network Functions**	**Score**
1	Cancer, endocrine system disorders, organismal injury, and abnormalities	42
2	Connective tissue disorders, developmental disorder, gastrointestinal disease	38
3	Embryonic development, organismal development, tissue morphology	33
4	Behavior, cell cycle, cell death and survival	31
5	Cardiovascular disease, organismal injury and abnormalities, molecular transport	31

**Table 2 bioengineering-11-01247-t002:** Top 5 of canonical pathways.

**NT vs. Pema**		
**Name**	***p*-Value**	**Overlap (%)**
CREB signaling in neurons	3.65 × 10^−9^	12.4
Cellular effects of sildenafil (Viagra)	3.69 × 10^−9^	11.9
Class A/1 (rhodopsin-like receptors)	4.36 × 10^−6^	15.1
Potassium channels	4.22 × 10^−8^	22.3
G-protein coupled receptor signaling	7.40 × 10^−7^	10.8
**NT vs. GW**		
**Name**	***p*-Value**	**Overlap (%)**
G alpha (s) signaling events	4.22 × 10^−5^	13.8
Class A/1 (rhodopsin-like receptors)	7.12 × 10^−5^	10.4
Activation of matrix metalloproteinases	1.84 × 10^−4^	24.2
Striated muscle contraction	3.51 × 10^−4^	22.2
Cardiac hypertrophy signaling (enhanced)	4.27 × 10^−4^	8.5

**Table 3 bioengineering-11-01247-t003:** DEGs commonly detected among different categories.

	Downregulation			Upregulation	
NT vs. Pema	NT vs. GW	Pema vs. GW	NT vs. Pema	NT vs. GW	Pema vs. GW
**H1-5**	**H1-5**		PKD1P1	PKD1P1	
NPEPPSP1	NPEPPSP1		SEPTIN7P3	SEPTIN7P3	
IKBKGP1	IKBKGP1			**HCLS1**	**HCLS1**
**ACTL10**	**ACTL10**			FDPSP2	FDPSP2
	**EFHC2**	**EFHC2**		EXOC5P1	EXOC5P1
	FRMPD2B	FRMPD2B		SLC4A1APP1	SLC4A1APP1
	**LINC00910**	**LINC00910**		TCAF2P1	TCAF2P1
	**PDE7B**	**PDE7B**		**MIR193BHG**	**MIR193BHG**
		**SNORA66**	**SNORA66**		
		ZNF890P	ZNF890P		
		**SMIM11**	**SMIM11**		
			**ANGPTL4**	**ANGPTL4**	**ANGPTL4**

## Data Availability

The data that support the findings of this study are available from the corresponding author upon reasonable request.

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
