# Peer review of "Pemafibrate Induces a Low Level of PPARα Agonist-Stimulated mRNA Expression of ANGPTL4 in ARPE19 Cell"

_bioengineering, 2024, doi:10.3390/bioengineering11121247_

Round 1

Reviewer 1 Report

Comments and Suggestions for Authors

The study “Pemafibrate induces a low level of PPARα agonist-stimulated mRNA expression of ANGPTL4 in ARPE19 cell” shows the novel mechanism of retinal ischemic diseases (RID) of the selective peroxisome proliferator-activated receptor α (PPARα) agonist (Pema) and the non-fibrate PPARα agonist (GW7647) in the ARPE19 cell line. I consider this an interesting and significant study that Pema can be applied in other clinical RID. Almost all of the limitations have been discussed by the authors, and I have a few questions as below:

1. The abbreviations for GW7647 and Pema should be consistent throughout the text. Please check.

2. For the DEGs of the 3 groups (table 3), it would be more convenient for readers to display them in a Venn plot.

The ANGPTL4 was confirmed as a target of the PPAR by many other studies. It’s should be a significant gene that induced by the 2 PPAR agonist in this study. I consider you should re-analyze the RNA-seq data with the fold-change ≥ 1.5 to show more potential mechanisms by the pema and GW7647. Additionally, the RNA-seq raw data should be submitted to public datasets, such as the GEO database.

3. Figure 3: The P value of ANGPTL4 with 10 uM in HepG2 should be checked. As the variation of the Pema is high.

4.For the discussion, you discussed more about the ANGPTL4 in this study. It’s better to show the relationship between the GO or PPI results and the metabolic functions you show in Figure 1.

Author Response

Dear Editor,

Thank you very much for the constructive comments concerning our manuscript “Pemafibrate induces a low level of PPARa agonist-stimulated mRNA expression of ANGPTL4 in ARPE19 cell”. We carefully checked all of the reviewers’ comments and prepared a revised version of our paper that takes these comments into account. The changes are listed below.

Reviewer 1 comments

The study “Pemafibrate induces a low level of PPARα agonist-stimulated mRNA expression of ANGPTL4 in ARPE19 cell” shows the novel mechanism of retinal ischemic diseases (RID) of the selective peroxisome proliferator-activated receptor α (PPARα) agonist (Pema) and the non-fibrate PPARα agonist (GW7647) in the ARPE19 cell line. I consider this an interesting and significant study that Pema can be applied in other clinical RID. Almost all of the limitations have been discussed by the authors, and I have a few questions as below:

  1. The abbreviations for GW7647 and Pema should be consistent throughout the text. Please check.

Answer; We sincerely appreciate your excellent comment. As pointed out, the abbreviations for GW7647 and Pema were consistently described throughout the text.

  1. For the DEGs of the 3 groups (table 3), it would be more convenient for readers to display them in a Venn plot.

Answer; We sincerely appreciate your excellent comment. As suggested as your excellent idea, a Venn plot was included as a new Fig. 3 to visualize more easily to understand table 3.

  1. The ANGPTL4 was confirmed as a target of the PPAR by many other studies. It’s should be a significant gene that induced by the 2 PPAR agonist in this study. I consider you should re-analyze the RNA-seq data with the fold-change ≥ 1.5 to show more potential mechanisms by the pema and GW7647. Additionally, the RNA-seq raw data should be submitted to public datasets, such as the GEO database.

Answer; We sincerely appreciate your excellent comment. As suggested, we have been carefully checked DEGs with the log2 fold-change ≥ ±1 and found several overlapped genes listed in Table 3 because of these DEGs may be related to PPARa related factors and others may be off-target factors. However, as another possibility, such off-target factors may influence ANGPTL4 expression, and thus we described to study these factors in our next project. This information is included in the last paragraph of discussion: ‘We acknowledge that the present study has several limitations. Firstly, the mechanisms underlying the different effects of Pema and GW on mRNA expression of ANGPTL4 and HIF1a as well as the difference in their effects in ARPE19 cells and HepG2 cells have not been elucidated despite the fact that Pema and GW were used under conditions of maximal and identical PPARa stimulation. This suggested that some unidentified off-targets effects specific to Pema but not GW may modulate PPARa-induced mRNA expression of ANGPTL4 and HIF1a. In fact, several DEGs in NT vs Pema, but not NT vs GW were identified in supplemental materials, and therefore additional study to re-analyze the RNA-seq data.’. In terms of our RNA sequencing data to be published in public data base, I wanted to do so, but in the present study, pemafibrate was supplied from Kowa company and they did not give us permission to do so because of some patent-related issue. Instead, we attached row data of DEGs in the supplemental materials, and thus, could you please understand this situation.   

  1. Figure 3: The P value of ANGPTL4 with 10 uM in HepG2 should be checked. As the variation of the Pema is high.

Answer; We sincerely appreciate your excellent comment. As pointed out, we carefully checked statistical analysis in Fig. 4.

  1. For the discussion, you discussed more about the ANGPTL4 in this study. It’s better to show the relationship between the GO or PPI results and the metabolic functions you show in Figure 1.

Answer; We sincerely appreciate your excellent comment. As suggested, additional discussion related to ANGPTL4 with the metabolic functions is added in second half of the 2nd paragraph of Discussion: ‘In addition, a previous study showed that overexpression of ANGPTL4 deteriorates mitochondrial functions in the liver tissues of db/db diabetic mice through downregulating numerous proteins associated with mitochondrial respiration [50].  Consistently to this report, the present study demonstrated that mitochondrial respiratory capacity in ARPE19 cells were significantly small by GW compared to Pema as shown in Fig. 1, presumably due to higher upregulation of ANGPTL4 by GW. Furthermore, baseline OCR/ECAR ratio suggested that metabolic dominance in glycolysis compared to mitochondrial function was more evident in order of non-treated control, Pema and GW. Since levels of mRNA expression of ANGPTL4 increased in that order, we speculated that ANGPTL4 may critically involved to regulate cellular metabolic functions in ARPE19 cells. Indeed, previous studies showed that ANGPTL4 enhances glycolysis in cancerous cells [51] [52]. Taken together, the excessive upregulation of ANGPTL4 in retinal pigment epithelial cells may be one of the pivotal mechanisms that increase the risk of retinal dysfunction via metabolic impairments.’.

Reviewer 2 comments

This is a straightforward study that reveals the different effects of PPARα agonists Pema and GW on ANGPTL4 of ARPE19 and HepG2 cells. The study is well-organized and the results are clearly presented. However, the overall manuscript needs to be carefully revised.

  1. Above all, it is well-known that Angptl4 is a PPARα and -γ target gene [Urvi Desai, et al. Proc Natl Acad Sci U S A. 2007], and previous studies have shown that PPARs function as nutritional sensors in hepatocytes and other tissues over 20 years ago. Angptl4 is not a “novel” target as the authors described in lines 311-312.

Answer; We sincerely appreciate your excellent comment. As suggested, corresponding sentence was corrected and reference ([Urvi Desai, et al. Proc Natl Acad Sci U S A. 2007]) was included as ref#33: ‘Among the ANGPTL family proteins, ANGPTL4 was simultaneously identified as a target for PPARa and PPARg, a fasting-induced factor from the liver, and an angiopoietin-related protein [33-36].’.

  1. Also, it is not surprising to see the upregulation of Angptl4 in PPARα-stimulated epithelium cells. Thus, the redundant content describing Angptl4 can be removed in the discussion section, but please emphasize the main finding: the different efficacy of Pema and GW for upregulating Angptl4.

Answer; We sincerely appreciate your excellent comment. As suggested, in the 1st paragraph of Discussion, lines 326-324 were removed and lines 336-340 was changed to ‘. In the current study, we first showed that the different efficacy of Pema and GW for upregulating ANGPTL4 in ARPE19 cells, that is, Pema-induced upregulation of ANGPTL4 was significantly less than that of GW, suggesting that Pema may be a preferable PPARa agonist to minimize ANGPTL4-related retinal risks.’.

  1. How about the expression levels of PPAR in retinal pigment epithelium compared with other tissues? The distribution/expression levels of PPAR in different organs or tissues would affect the drug efficacy significantly.

Answer; We sincerely appreciate your excellent comment. I agree that the distribution/expression levels of PPAR in different organs or tissues would affect the drug efficacy significantly, and thus, difference of the expression levels of PPAR in retinal pigment epithelium and other tissues should be very important. In the present study, we compared 2 different PPARa agonists, Pema and GW on ARPE19 cells or HepG2 cells. Therefore, as suggested, Pema and GW induced different ANGPTL4 expression profiles between ARPE19 and HepG2 cells. However, we emphasized different ANGPTL4 expression profiles between Pema and GW even though in ARPE19 cells with an identical level of PPARa expression. 

  1. An untreated group needs to be included as a negative control/baseline for the expression of ANGPTL4 and HIF1α in PPAR agonist-treated cells.

Answer; We sincerely appreciate your excellent comment. As pointed out, a negative control/baseline for the expression of ANGPTL4 and HIF1α in PPAR agonist-treated cells were include in Fig. 4.

Reviewer 2 Report

Comments and Suggestions for Authors

This is a straightforward study that reveals the different effects of PPARα agonists Pema and GW on ANGPTL4 of ARPE19 and HepG2 cells. The study is well-organized and the results are clearly presented. However, the overall manuscript needs to be carefully revised.

Above all, it is well-known that Angptl4 is a PPARα and -γ target gene [Urvi Desai, et al. Proc Natl Acad Sci U S A. 2007], and previous studies have shown that PPARs function as nutritional sensors in hepatocytes and other tissues over 20 years ago. Angptl4 is not a “novel” target as the authors described in lines 311-312.

Also, it is not surprising to see the upregulation of Angptl4 in PPARα-stimulated epithelium cells. Thus, the redundant content describing Angptl4 can be removed in the discussion section, but please emphasize the main finding: the different efficacy of Pema and GW for upregulating Angptl4.

How about the expression levels of PPAR in retinal pigment epithelium compared with other tissues? The distribution/expression levels of PPAR in different organs or tissues would affect the drug efficacy significantly.

An untreated group needs to be included as a negative control/baseline for the expression of ANGPTL4 and HIF1α in PPAR agonist-treated cells.

Author Response

(The authors gave the same response as above.)

Round 2

Reviewer 1 Report

Comments and Suggestions for Authors

Thank you for the response.

Author Response

Dear Editor,

Thank you very much for the constructive comments concerning our manuscript “ROCK inhibitors significantly modulate the epithelial mesenchymal transition induced by TGF-β2 in the 2-D and 3-D cultures of human corneal stroma fibroblasts”. As pointed out by both reviewers, English is carefully checked by a native English speaking scientist, Dr. Stewart Chisholm, and correction are highlighted by yellow. 

Reviewer 2 Report

Comments and Suggestions for Authors

The comments have been addressed appropriately.

Minor language edits are needed before publication.

Author Response

(The authors gave the same response as above.)
